# The short-chain fatty acid pentanoate suppresses autoimmunity by modulating the metabolic-epigenetic crosstalk in lymphocytes

Maik Luu[1], Sabine Pautz[1,2], Vanessa Kohl[1], Rajeev Singh[3], Rossana Romero[1], Sébastien Lucas[4], Jörg Hofmann[5], Hartmann Raifer[6], Niyati Vachharajani[1,7], Lucia Campos Carrascosa[1], Boris Lamp[8], Andrea Nist[8], Thorsten Stiewe [8,9], Yoav Shaul[10], Till Adhikary[3], Mario M. Zaiss[4], Matthias Lauth[3], Ulrich Steinhoff[1] & Alexander Visekruna [1]

Short-chain fatty acids (SCFAs) have immunomodulatory effects, but the underlying mechanisms are not well understood. Here we show that pentanoate, a physiologically abundant SCFA, is a potent regulator of immunometabolism. Pentanoate induces IL-10 production in lymphocytes by reprogramming their metabolic activity towards elevated glucose oxidation. Mechanistically, this reprogramming is mediated by supplying additional pentanoate-originated acetyl-CoA for histone acetyltransferases, and by pentanoate-triggered enhancement of mTOR activity. In experimental mouse models of colitis and multiple sclerosis, pentanoate-induced regulatory B cells mediate protection from auto-immune pathology. Additionally, pentanoate shows a potent histone deacetylase-inhibitory activity in CD4$^+$ T cells, thereby reducing their IL-17A production. In germ-free mice mono-colonized with segmented filamentous bacteria (SFB), pentanoate inhibits the generation of small-intestinal Th17 cells and ameliorates SFB-promoted inflammation in the central nervous system. Taken together, by enhancing IL-10 production and suppressing Th17 cells, the SCFA pentanoate might be of therapeutic relevance for inflammatory and autoimmune diseases.

[1] Institute for Medical Microbiology and Hygiene, Philipps-University Marburg, Marburg 35043, Germany. [2] Department of Biochemistry, University of Kassel, Kassel 34132, Germany. [3] Institute of Molecular Biology and Tumor Research, Center for Tumor- and Immunobiology, Philipps-University Marburg, Marburg 35043, Germany. [4] Department of Internal Medicine 3, Rheumatology and Immunology, Friedrich-Alexander-University-Erlangen-Nürnberg and Universitätsklinikum Erlangen, Erlangen 91054, Germany. [5] Division of Biochemistry, Department of Biology, Friedrich-Alexander-University-Erlangen-Nürnberg, Erlangen 91054, Germany. [6] Flow Cytometry Core Facility, Philipps-University Marburg, Marburg 35043, Germany. [7] Department of Medicine, Division of Gastroenterology, Vanderbilt University Medical Center, Nashville 37232 TN, USA. [8] Genomics Core Facility, Philipps-University Marburg, Marburg 35043, Germany. [9] Institute of Molecular Oncology, Philipps-University Marburg, Marburg 35043, Germany. [10] Department of Biochemistry and Molecular Biology, The Institute for Medical Research Israel-Canada, The Hebrew University-Hadassah Medical School, Jerusalem 9112102, Israel. Correspondence and requests for materials should be addressed to A.V. (email: alexander.visekruna@staff.uni-marburg.de)

Short-chain fatty acids (SCFAs) such as acetate (C2), propionate (C3), and butyrate (C4) are generated by bacterial fermentation of dietary fiber in the intestinal lumen[1]. Soluble microbial factors including SCFAs act as important signals physically bridging the gap between the commensal microbiota and mucosal immune system[2–4]. SCFAs have been shown to induce the differentiation of colonic regulatory T cells (Tregs) and to enhance the gut barrier function[5–8]. The impact of SCFAs on Tregs was suggested to be mediated via SCFA-receptor FFAR2 (GPR43) and histone deacetylase (HDAC)-inhibitory activity[5,8]. SCFAs are not only able to protect from mucosal inflammation and colorectal tumorigenesis, but may also act in a systemic manner to ameliorate T cell-driven autoimmunity in the brain and allergic asthma in the lung[9–11]. Moreover, butyrate has been recently shown to mitigate graft-versus-host disease in mice[12]. It has been suggested that medical food containing SCFAs might counter severe immunological defects as feeding mice a combined acetate- and butyrate-yielding diet provides complete protection against type 1 diabetes in mice[13]. Thus, gut microbiota-derived metabolites might be of therapeutic benefit to several immunological disorders. Notably, not only beneficial but also unfavorable effects of SCFAs on our health have been described. The SCFA formate (C1) impacts directly on pathogens to upregulate the expression of invasion genes during the infection with *Salmonella typhimurium*. Similarly, butyrate has been shown to act on the locus of enterocyte effacement (LEE) pathogenicity island of EHEC, which enables this pathogen to efficiently colonize the host epithelium[14]. Furthermore, SCFAs are capable of inducing tissue-specific inflammation in the ureter and kidney, leading to T cell-mediated renal disease with progressive ureteritis and kidney hydronephrosis[15]. Thus, although SCFAs promote our health at steady state by increasing intestinal epithelial cell integrity, pathogen-specific antibody responses and the numbers of colonic Tregs, they are also potentially able to boost inflammatory responses mediated by epithelial cells or T lymphocytes[16–19].

Here, we wish to evaluate the capacity of the SCFA pentanoate (valerate, C5) as a potential therapeutic for autoimmune and inflammatory diseases. We identify pentanoate as a potent and inexpensive immunomodulatory molecule, being able to suppress aberrant immune cell activation in the gut and central nervous system (CNS). Our study reveals previously unknown metabolic rewiring caused by pentanoate on mTOR activity in B cells and CD4[+] effector T lymphocytes, leading to increased glucose oxidation, elevated acetyl-CoA levels, and strong secretion of the immunosuppressive cytokine IL-10. In effector T cells, the HDAC-inhibitory activity of pentanoate suppresses the expression of IL-17A, leading to the amelioration of autoimmune inflammation in the CNS. These data provide insights into the mechanisms by which SCFAs regulate the balance between tolerance and adverse immune responses.

## Results

**Pentanoate suppresses IL-17A production**. We first examined the abundance of SCFAs in the murine gut lumen by ultra-high pressure liquid chromatography-mass spectrometry (UHPLC-MS). We observed that not only the abundantly produced SCFAs acetate, propionate, and butyrate, but also pentanoate is likely the product of bacterial fermentation as it was present in the gut lumen of wild-type (WT) but not in that of GF mice (Supplementary Fig. 1a). The fecal concentration of pentanoate was much lower than that of acetate, propionate, and butyrate (Supplementary Fig. 1b), suggesting that pentanoate might be generated by fermentation of different dietary components. *Prevotella* species are potent SCFA-producers exclusive to the humans with high fiber intake[20–22]. Gas chromatography (GC)-MS analysis

revealed that *Prevotella* generated predominantly acetate without detectible levels of pentanoate (Supplementary Fig. 1c). Thus, the low intestinal pentanoate production is likely not dependent on bacterial fermentation of dietary fiber.

While an increasing body of evidence suggests an immunomodulatory activity for acetate, propionate, and butyrate, the role for the SCFA pentanoate in regulating the immune cell function is still unknown. To explore a potential therapeutic capacity of pentanoate, we generated pathogenic Th17 cells with IL-6 and IL-23 in combination with IL-1β. After 3 days of differentiation, pentanoate treatment effectively inhibited the proliferation of Th17 lymphocytes and their IL-17A production (Fig. 1a). The global RNA-seq analysis revealed that pentanoate upregulated *Il10* expression and downregulated most of the Th17-associated genes including *Rorc*, *Il21*, *Stat3*, and predominantly *Tgfb3*, which is endogenously produced by pathogenic Th17 cells (Fig. 1b). TGF-β3 essentially contributes to the maintenance of the pathogenic phenotype of Th17 cells by upregulating the IL-23 receptor[23]. During the course of experimental autoimmune encephalomyelitis (EAE), Th17 cells have been described as a central pathogenic T cell population contributing to the development of autoimmune inflammation in the CNS[24,25]. Development of highly pathogenic Th17 cells has been reported to be essential for robust EAE development[23,26]. To investigate whether pentanoate could modulate inflammation of the CNS, we induced EAE in *FIR × tiger* reporter mice. The treatment of mice with pentanoate ameliorated EAE severity and reduced the number of infiltrating CD4[+] and CD8[+] T cells in the CNS (Supplementary Fig. 2a, b). Pentanoate-treated mice exhibited low frequencies of IL-17A[+] and IFN-γ[+]IL-17A[+] cells within CD4[+] and CD8[+] T lymphocytes (Supplementary Fig. 2c-e). Previously, the presence of IL-17A[+] and IFN-γ[+] Tregs in the inflamed CNS was described, questioning their anti-inflammatory nature in this highly inflamed environment[27]. We found that during EAE development, a significant proportion of Foxp3[+] (RFP[+]) Tregs in the inflamed CNS of *FIR × tiger* mice co-expressed IFN-γ and IL-17A. Although in vivo pentanoate treatment did not alter the frequency of Tregs in the CNS, it strongly reduced the proportion of IL-17A[+] and IFN-γ[+]IL-17A[+] cells within the Foxp3[+] Treg population (Supplementary Fig. 2f-h).

We next tested the hypothesis that pentanoate might be able to suppress the generation of intestinal Th17 cells in the context of nonpathogenic bacteria–immune system interaction, which could modulate the progression of EAE in mice. Segmented filamentous bacteria (SFB), commensal gut bacteria of mammals, induce the expansion of Th17 cells in murine small intestine[28]. While GF mice are highly resistant to the EAE development[29], mono-colonization of these animals with SFB induces pathogenic Th17 responses and promotes the CNS inflammation[30] (Fig. 1c, d). The treatment of SFB-mono-colonized GF mice with pentanoate significantly ameliorated EAE and led to reduced cell numbers and frequencies of Th17 cells in the CNS (Fig. 1c–f). Remarkably, the flow cytometry analysis revealed that during EAE development, pentanoate was able to inhibit the generation of intestinal Th17 cells in GF mice mono-colonized with SFB (Fig. 1d, f).

**Pentanoate induces IL-10 expression in effector T cells**. Butyrate and propionate influence the epigenetic status of immune cells by inhibiting the enzymatic activity of HDACs[5,31]. To study if pentanoate is able to modulate the function of these enzymes, we measured the HDAC activity in murine CD4[+] T lymphocytes in the presence of various SCFAs. While butyrate, pentanoate, and propionate exerted a potent HDAC-inhibitory activity, acetate and hexanoate (caproate, C6) almost completely lacked HDAC-inhibitory properties (Fig. 2a). The potent HDAC-

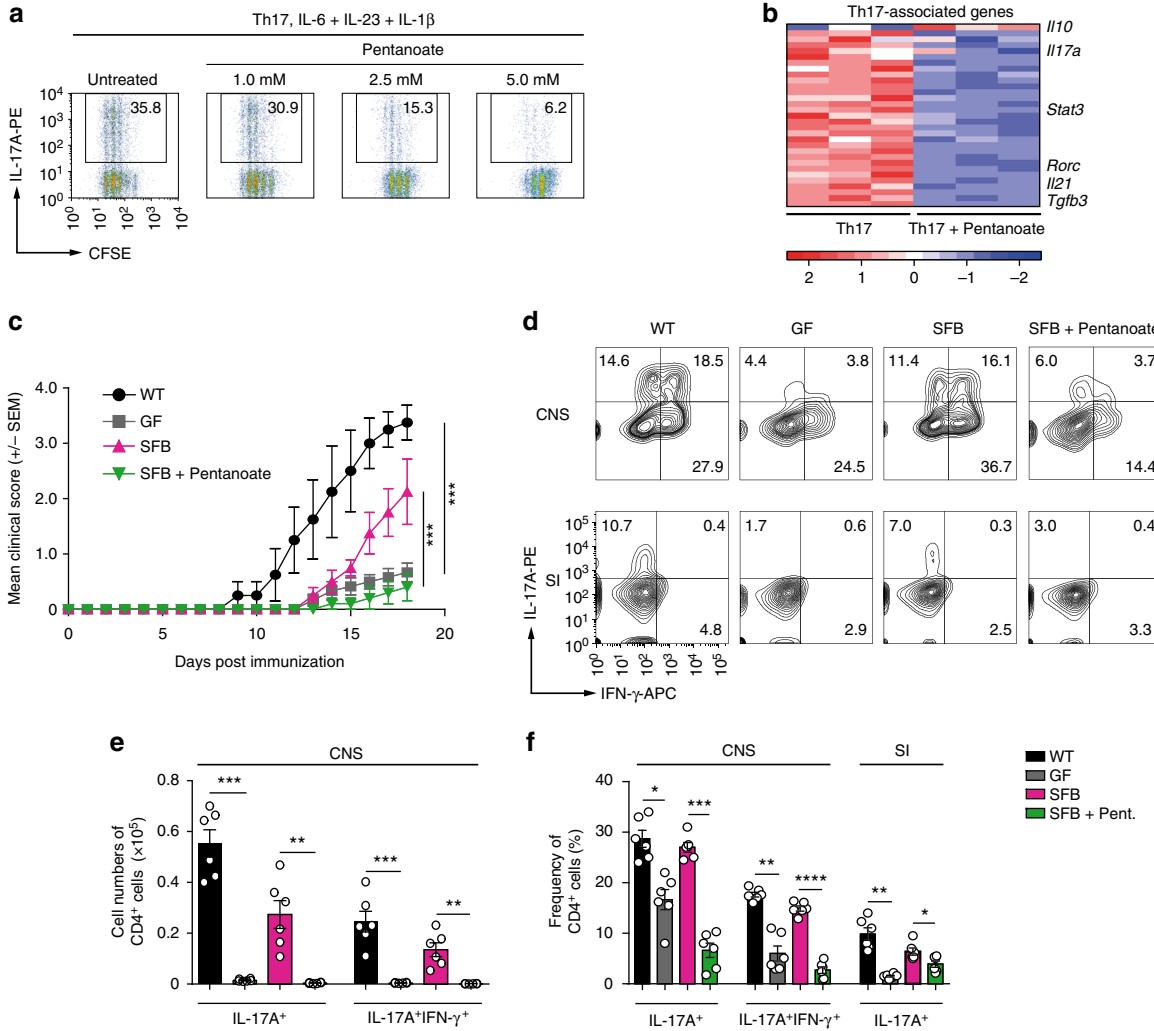

**Fig. 1** Pentanoate inhibits induction of IL-17A. **a** Pathogenic Th17 cells were generated by polarizing CD4+ T cells in the presence of IL-6, IL-23, and IL-1β and increasing pentanoate concentrations. Staining for IL-17A and CFSE is shown as a representative of three similar experiments. **b** RNA-seq analysis of pathogenic Th17 cells in the presence or absence of pentanoate. Heatmap of downregulated Th17-associated genes is shown. The FPKM values were z-transformed and plotted. **c** Clinical EAE scores for WT, GF, GF + SFB and GF + SFB mice treated with pentanoate as described in Methods. Mice were immunized with MOG peptide emulsified in CFA + pertussis toxin ($n = 6$ animals per group from one representative experiment of two). Mice were monitored daily for the progression of EAE. **d** Representative contour plots showing the frequency of IFN-γ+ and IL-17A+CD4+ in the central nervous system (CNS) and small intestine (SI) among different mice groups on day 18 after EAE induction. **e** Absolute cell numbers of IL-17A+ and IL-17A+ IFN-γ+CD4+ T cells in four group of mice treated as described in (**c**) and analyzed on day 18 after EAE induction. **f** Percentages of IL-17A+ (CNS and SI) and IL-17A+IFN-γ+CD4+ T cells (CNS) in WT, GF, GF + SFB mice, and GF + SFB animals treated with pentanoate on day 18 after EAE induction. Error bars indicate SEM. *$P < 0.05$, **$P < 0.01$, ***$P < 0.001$ (one-way ANOVA test)

inhibitory activity of pentanoate was likely involved in the reduction of IL-17A expression as the treatment of Th17 cells with a pan-HDAC inhibitor trichostatin A (TSA) caused a similar effect (Fig. 2b). Dietary fiber-dependent SCFAs were shown to enhance the IL-10 expression in mucosal T cells[8]. Interestingly, we observed that TSA was not able to induce the production of IL-10 (GFP+) by Th17 cells derived from *FIR × tiger* mice. To explore whether SCFA-mediated metabolic alterations might regulate the balance between pro- and anti-inflammatory cytokines, we examined the impact of pentanoate in the presence of 2-deoxy-D-glucose (2-DG, an inhibitor of glycolysis) on concomitant expression of IL-17A and IL-10 in Th17 cells. Of note, the frequency of IL-10+ (GFP+) cells was strongly increased after stimulation of Th17 cells with pentanoate. The co-treatment of Th17 cells with 2-DG led to a complete blockade of pentanoate-mediated IL-10 induction, suggesting that enhanced glycolysis

but not HDAC inhibitory activity of pentanoate was responsible for this effect (Fig. 2b–d). Indeed, we found that Th17 cells treated with pentanoate were able to increase the extracellular acidification rate (ECAR) and enhance their glycolytic activity (Fig. 2e). Thus, the enhancement of glycolysis, mediated by pentanoate (increased IL-10 production) together with its HDAC-inhibitory activity (reduced IL-17A expression), leads to metabolic and epigenetic reprogramming and loss of the pathogenic phenotype of Th17 cells.

AKT/mTOR signaling pathway, which is a crucial regulator of cellular processes in response to environmental cues, controls the glycolytic metabolism in cancer and other cells[32,33]. We were wondering whether pentanoate is capable of increasing the activity of mTOR complex in Th17 cells. Indeed, the activity of both, mTOR and its target molecules, was enhanced after treatment of Th17 cells with pentanoate (Fig. 2f). AMP-activated

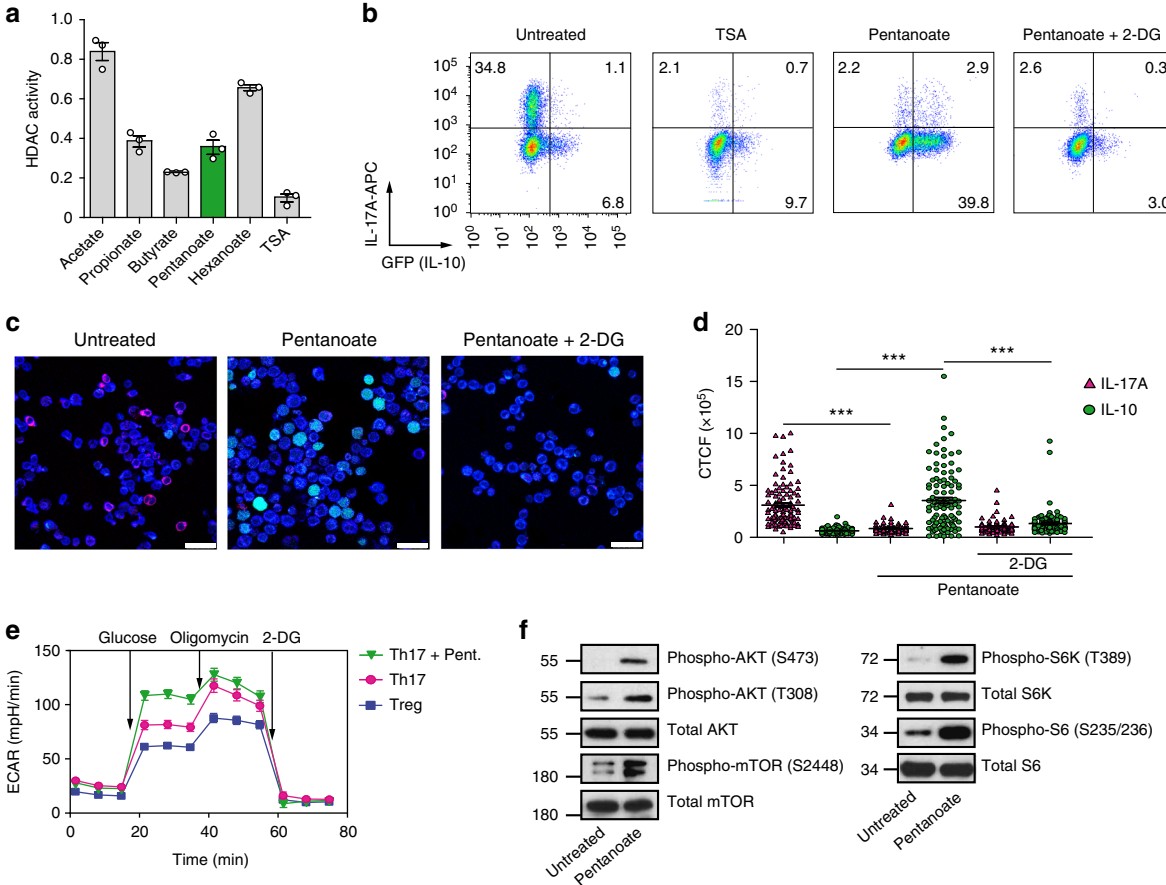

**Fig. 2** Pentanoate induces IL-10 expression in Th17 cells. **a** Bar graphs show the HDAC activity in whole cell lysates obtained from murine CD4+ T cells treated with SCFAs (5 mM) or TSA (500 nM). The HDAC activity of untreated CD4+ T cells is arbitrary set to 1 (n = 3). **b–d** CD4+ T cells isolated from *FIR x tiger* mice were polarized under Th17-inducing conditions and treated with TSA (10 nM), pentanoate (5 mM) or pentanoate + 2-DG (2 mM). Cells were analyzed by flow cytometry (**b**) and fluorescence microscopy (**c**, **d**, scale bar = 25 micron) for CD4 (blue), IL-17A (red) and GFP (green) expression on day 4 of the cell culture. For counting of cytokine-positive cells, the CTCF method was used as described in Methods. ***P < 0.001 (Student's t-test). **e** CD4+ T cells isolated from spleens and LNs were polarized under Th17- or Treg-inducing conditions for 3 days. A representative ECAR measurement of Th17 cells polarized in the presence and absence of pentanoate is shown (Tregs served as control cells). Three independent experiments were performed. **f** Detection of the main components of the AKT/mTOR signaling pathway by immunoblotting of Th17 cell lysates after treatment with pentanoate. Three similar experiments were performed

protein kinase (AMPK) activity is known to block the activation of mTOR pathway[16]. Accordingly, inhibition of AKT/mTOR signaling (by rapamycin or AKT-inhibitor IV) but also AMPK activation by metformin substantially suppressed IL-10 production in pentanoate-treated Th17 cells (Fig. 3a, b). Further, the cellular levels of ATP were significantly increased after treatment of Th17 lymphocytes with pentanoate (Fig. 3c). Thus, pentanoate likely inhibits the activity of AMPK by elevating ATP and depleting AMP levels in the cells. Importantly, only pentanoate-treated but not control Th17 cells increased their IL-10 production following glucose supplementation in the presence of glutamine (Fig. 3d). Increasing glutamine concentrations were not able to further increase the secretion of IL-10 in the presence of glucose (Fig. 3e). These data suggest that enhanced glycolytic metabolism (mediated by pentanoate), but not glutaminolysis, controls increased IL-10 expression in Th17 cells. Our findings suggest that pentanoate is likely converted into acetyl-CoA as pentanoate-treated Th17 cells exhibit strongly elevated acetyl-CoA concentration (Fig. 3f). Recently, glycolysis-derived pyruvate oxidation and the subsequent conversion of citrate to acetyl-CoA, mediated by the enzyme ATP citrate-lyase (ACLY), was shown to be crucial for rapid IFN-γ generation in memory CD8+ T cells[34].

Our findings suggest that both the inhibition of pyruvate kinase-M2 (shikonin), which catalyzes the conversion from phosphoenolpyruvate to pyruvate, and that of ACLY (SB204990) leads to decreased IL-10 secretion by pentanoate-treated Th17 cells (Fig. 3g). In line with these observations, we found that shifting glucose metabolism from aerobic glycolysis (lactate production) to pyruvate oxidation by inhibiting pyruvate dehydrogenase kinase (PDHK) and activating pyruvate dehydrogenase (PDH) with dichloroacetate (DCA) resulted in elevated IL-10 production in Th17 cells even in the absence of pentanoate. This effect was reversed by the addition of 2-DG in the cell culture (Fig. 3h). Acetyl-CoA is the only source for histone acetyltransferase (HAT)-mediated acetylation of histones in the cell[35]. Of note, we observed increased acetylation of the histone H4 at the *Il10* promotor after treatment of Th17 cells with pentanoate (Fig. 3i). As TSA treatment of Th17 cells was not able to strongly induce IL-10 expression, we suggest that pentanoate, although a potent HDAC inhibitor, might rather influence the net H4 acetylation by serving as a metabolic input for the HAT substrate acetyl-CoA. Accordingly, we observed that increasing concentrations of acetate (a very weak HDAC inhibitor) elevated IL-10 secretion by Th17 cells, likely by acting as a precursor molecule for acetyl-

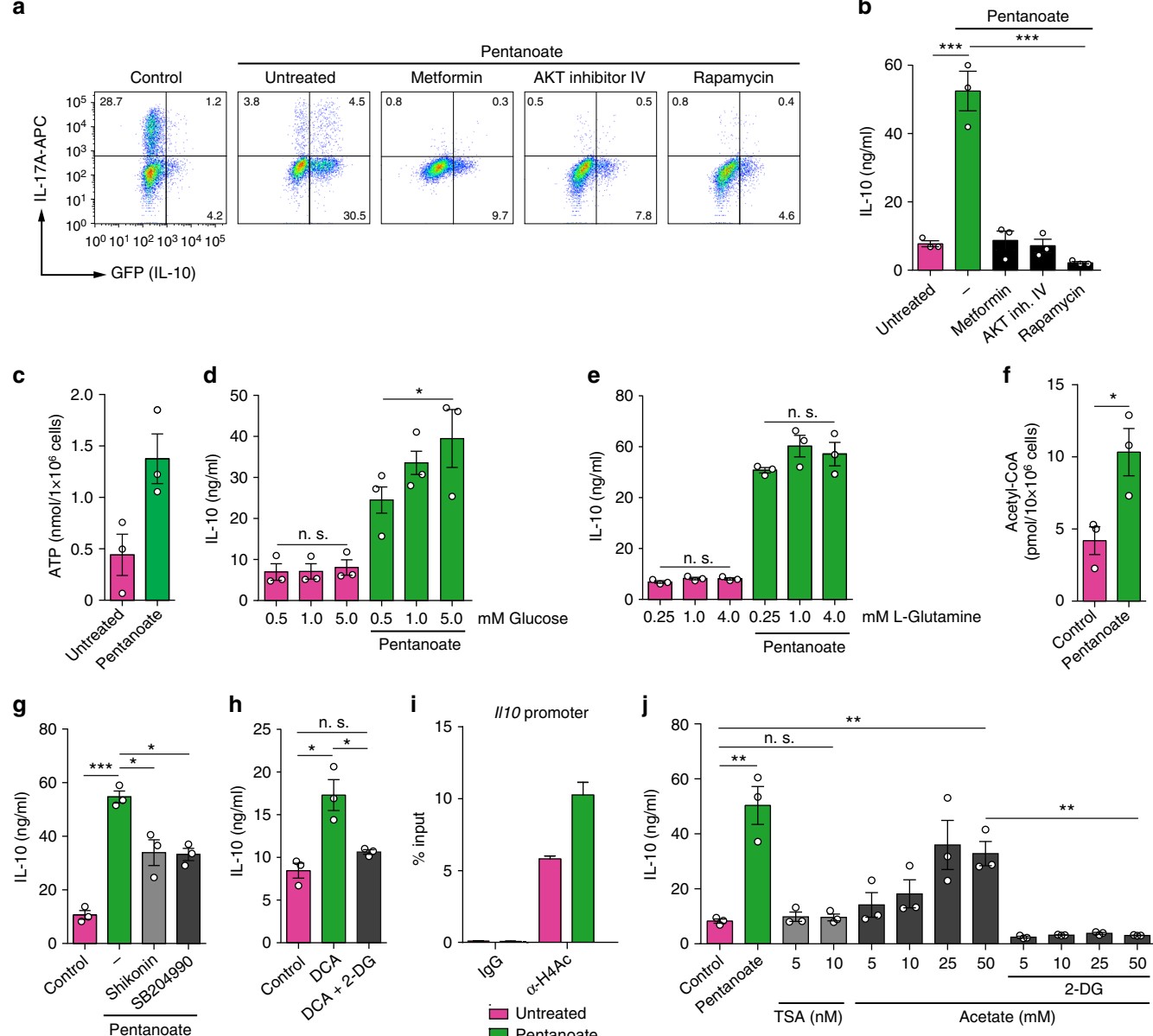

**Fig. 3** Pentanoate-mediated glucose oxidation is linked to induction of IL-10 expression. **a**, **b** Th17 cells purified and generated from the spleens and LNs of *FIR x tiger* mice were treated with AMPK activator (metformin) or AKT (AKT inhibitor IV) and mTOR (rapamycin) inhibitors in the presence of pentanoate. The frequency of IL-10+ and IL-17A+ cells within Th17 lymphocytes was analyzed by flow cytometry (**a**). The secretion of IL-10 was determined by ELISA (**b**). Three experiments were performed. **c** ATP production by Th17 cells in the absence or presence of pentanoate (2.5 mM). Three experiments were performed. **d**, **e** Th17 cells were treated with 2.5 mM pentanoate in the presence of increasing concentrations of glucose (**d**) or L-Glutamine (**e**). Three experiments were performed. **f** Cellular acetyl-CoA concentrations in pentanoate-treated Th17 cells were measured using fluorometric acetyl-CoA assay kit. **g** IL-10 secretion by pentanoate-treated Th17 cells in the presence of PKM2 (shikonin) or ACLY (SB204990) inhibitors was measured by ELISA ($n = 3$). **h** Th17 cells were treated with DCA in the presence or absence of 2-DG for 3 days. IL-10 secretion was determined by ELISA ($n = 3$). **i** ChIP analysis of acetylated H4 at the *Il10* promoter in pentanoate-treated Th17 cells. Three similar experiments were performed. **j** IL-10 secretion by Th17 cells cultured for 3 days upon TSA or acetate treatment. Pentanoate-treated Th17 cells served as a positive control for ELISA measurement. Acetate-treated cells were additionally treated with 2-DG (1 mM). Error bars indicate SEM. n.s. = not significant, *$P < 0.05$, **$P < 0.01$, ***$P < 0.001$ (Student's *t*-test)

CoA. Finally, the presence of 2-DG resulted in a blockade of acetate-mediated IL-10 secretion by Th17 cells, strongly supporting our hypothesis (Fig. 3j).

**Pentanoate does not have an impact on Tregs.** It was shown that in mice fed a HFD or being orally treated with fiber-dependent SCFAs the differentiation of intestinal Tregs was induced[5,6]. We next examined whether pentanoate might also

potentiate Treg differentiation by inducing epigenetic modifications at the *Foxp3* locus. Surprisingly, only butyrate increased the frequency of inducible Foxp3+ Tregs in vitro (Supplementary Fig. 3a, b). Furthermore, the oral treatment of GF mice revealed that only butyrate but not pentanoate increased the cell number and frequency of mucosal Tregs (Supplementary Fig. 3c-e). To further assess the impact of pentanoate on Treg function, *FIR × tiger* reporter mice were orally treated with butyrate or pentanoate for 4 weeks and the frequency of intestinal Foxp3+ Tregs

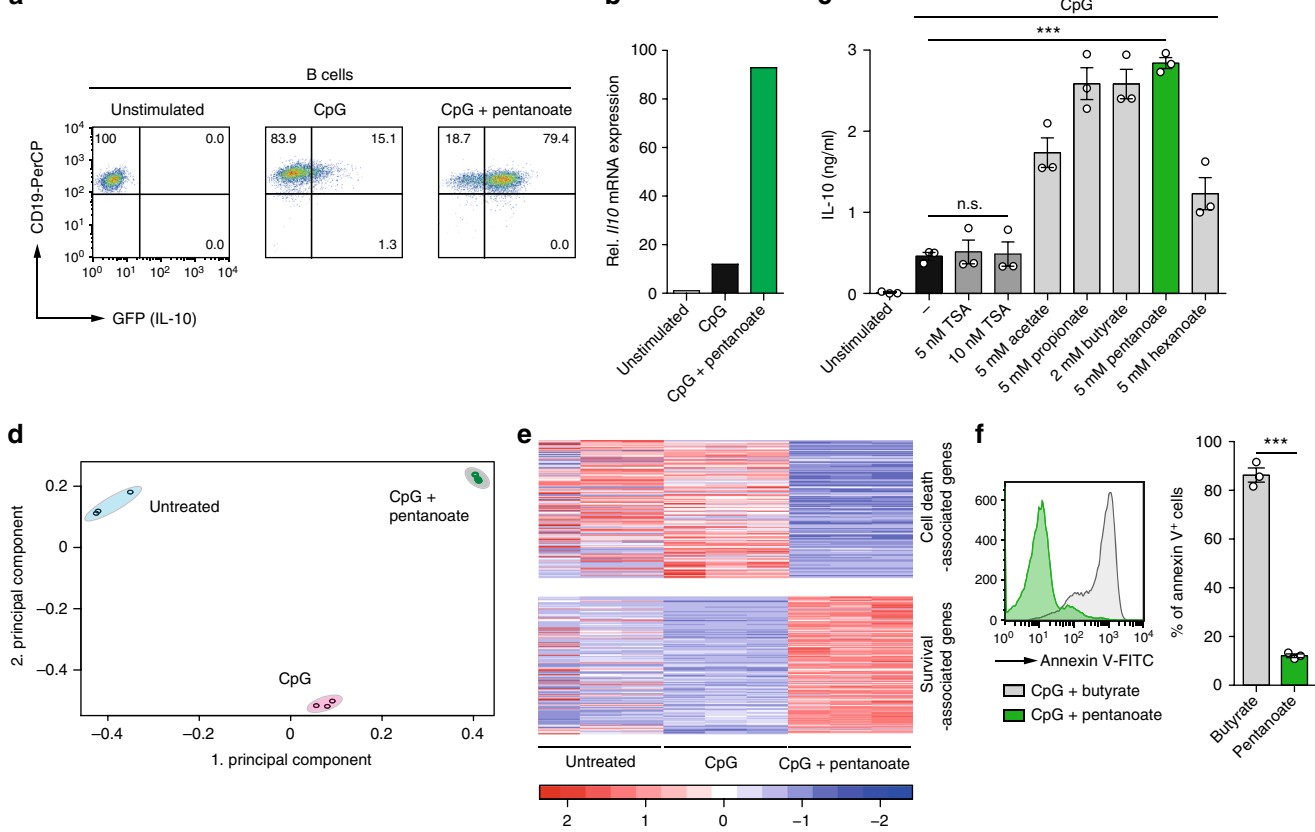

**Fig. 4** Pentanoate enhances IL-10 expression in Bregs. **a** Splenic B cells purified from spleens of *FIR × tiger* mice were activated with CpG in the absence or presence of pentanoate. Representative dot plots show the frequencies of IL-10+ B cells on day 4 of the cell culture. **b** Representative RT-qPCR analysis of *Il10* expression in Bregs treated with pentanoate for 3 days (one of three similar experiment is shown). **c** Bar graphs indicate secreted amounts of IL-10 by Bregs incubated for 3 days with various SCFAs or with TSA. Three experiments were performed. **d**, **e** B cells were treated as described in (**a**) and analyzed via RNA-seq. PCA (**d**) was performed on FPKM values. Heatmap (**e**) of cell death- and survival-associated genes, regulated upon pentanoate treatment, and overlapping with gene sets based on the IPA-database are shown. FPKM values of overlapping genes were z-transformed and plotted. **f** Histogram (left) and bar graphs (right) showing the frequency of Annexin V+ Bregs treated with SCFAs pentanoate (5 mM) or butyrate (5 mM) for 24 h. Three independent experiments were performed. Error bars indicate SEM. n.s. = not significant, **$P < 0.01$, ***$P < 0.001$ (Student's *t*-test)

was measured by flow cytometry. Consistent with in vitro observations, butyrate but not pentanoate was able to enhance the percentage of colonic Tregs (Supplementary Fig. 3f-h). Interestingly, although pentanoate did not promote Treg expansion, it exhibited similar effects to butyrate on IL-10-expressing Treg subset in the colon. Both SCFAs significantly expanded the IL-10+ subpopulation within colonic Tregs as compared to untreated mice (Supplementary Fig. 3i, j). Taken together, although pentanoate treatment induces epigenetic changes in Tregs, it cannot mimic the effects of butyrate, which was shown to boost Treg expansion by enhancing acetylation of the histones at the *Foxp3* locus[5,6].

**Pentanoate-treated B cells upregulate IL-10 secretion**. Since microbiota-derived signal molecules simultaneously affect various immune cell populations, we next examined whether SCFA treatment also impacts IL-10 production in regulatory B cells (Bregs). Some B lymphocytes are able to suppress inflammation through the secretion of IL-10 instead of producing antibodies[36]. We found that splenic B cells stimulated either with CpG or LPS (Bregs) strongly enhanced their IL-10 expression in the presence of pentanoate (Fig. 4a, b and Supplementary Fig. 4a, b). Butyrate, propionate, and acetate also potentiated IL-10 production in Bregs. Similar to Th17 cells, TSA was not capable of strongly

increasing IL-10 secretion in Bregs (Fig. 4c). To better understand the global effects of pentanoate on Bregs, we performed RNA-seq on CpG-induced Bregs with or without pentanoate treatment. Principal component analysis performed on RNA-seq data revealed a strong difference between Bregs generated in the presence and absence of pentanoate (Fig. 4d). Interestingly, by performing hypergeometric enrichment analysis on the genes that were regulated by pentanoate treatment of Bregs, we found a significant overlap with cell death- and survival-associated genes. Several genes related to apoptosis were downregulated upon pentanoate treatment, while genes associated with pro-survival properties were upregulated (Fig. 4e). Flow cytometry of Annexin V+ apoptotic cells revealed that among four different SCFAs, pentanoate was the only one, which was not only able to increase secretion of IL-10 but also to significantly suppress apoptosis in Bregs (Supplementary Fig. 4c, d). Remarkably, in contrast to pentanoate, which strongly reduced cell death of Bregs, the same concentration of butyrate promoted the apoptosis of Bregs to nearly 100% (Fig. 4f and Supplementary Fig. 4c). To examine the role of pentanoate on in vivo functionality of Bregs, naive CD4+ T cells alone or together with pentanoate-treated Bregs were adoptively transferred into Rag1-deficient mice. Mice reconstituted with naive T cells showed a progressive weight loss and colitis development by day 50, while the prevention of weight loss, amelioration of immunopathology, and reduced CD4+

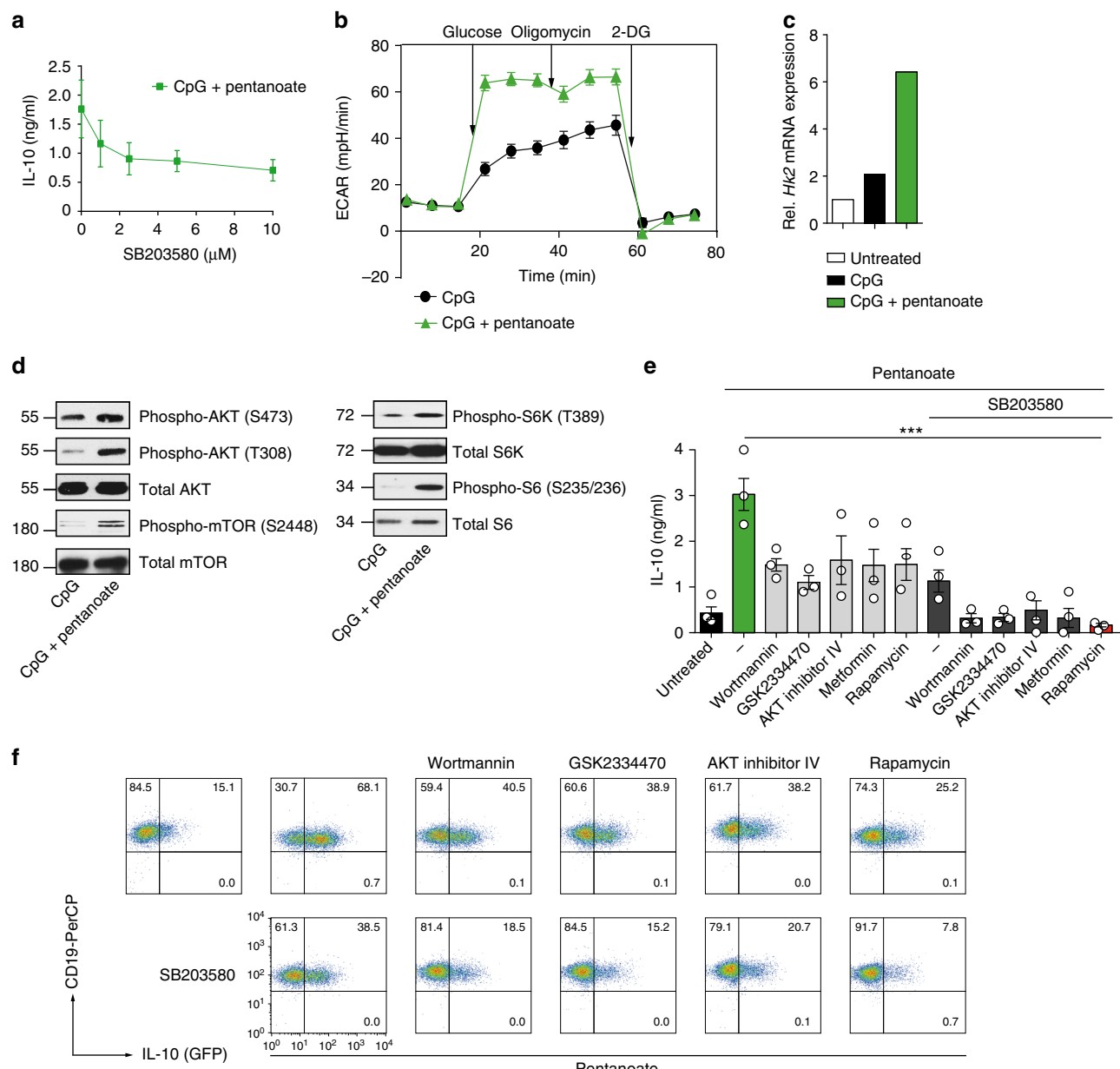

**Fig. 5** Pentanoate-induced IL-10 expression in Bregs is dependent on glycolysis and p38 MAPK activity. **a** Bregs were generated with 4 µg/ml CpG and 5 mM pentanoate in presence of increasing amounts of p38 inhibitor SB203580. IL-10 secretion was determined by ELISA. Three experiments were performed. **b** ECAR measurement of CpG-induced Bregs generated in the presence and absence of pentanoate. Three independent experiments were performed. **c** Bar graphs show *Hk2* expression in pentanoate-treated Bregs on day 3 of the cell culture. One of three similar experiments is shown. **d** Immunoblot analysis of main components of AKT/mTOR signaling pathway in murine Bregs in the presence of pentanoate (5 mM). Three similar experiments were performed. **e**, **f** ELISA (**e**) and FACS analysis (**f**) of CD19⁺IL-10⁺ Bregs generated with CpG and pentanoate in the presence of AMPK activator (metformin) or PI3K (wortmannin), PDK1 (GSK2334470), mTOR (rapamycin), and AKT (AKT inhibitor IV) inhibitors with or without SB203580. Three experiments were carried out. Error bars indicate SEM. ***$P < 0.001$ (Student's $t$-test)

effector T cell numbers in colonic lamina propria and mesenteric lymph nodes (mLN) were observed in the presence of pentanoate-treated Bregs (Supplementary Fig. 4e, f). These data suggest that pentanoate but not butyrate might be therapeutically exploited for stabilizing suppressive phenotype of Bregs.

**Pentanoate-treated Bregs inhibit EAE development in mice.** In B cells, the activation of p38 mitogen-activated protein kinase (MAPK) signaling pathway leads to the induction of IL-10 expression[37]. To test a possible influence of p38 on the pentanoate-induced secretion of IL-10, we treated Bregs with

CpG and pentanoate in the presence of increasing concentrations of p38 inhibitor SB203580. After blocking the activation of p38 kinase, pentanoate-generated Bregs still produced IL-10 to some extent, indicating that additional mechanisms are involved in the pentanoate-mediated modulation of Bregs (Fig. 5a). Interestingly, many of the genes altered upon pentanoate treatment were related to metabolic pathways (Supplementary Fig. 5a, a complete list of metabolically-related genes was reported by us previously[38]). To test if pentanoate is capable of influencing glycolysis in B cells, we examined the ECAR in CpG-generated Bregs in the presence or absence of pentanoate. We observed that pentanoate treatment increased both glycolysis and glycolytic

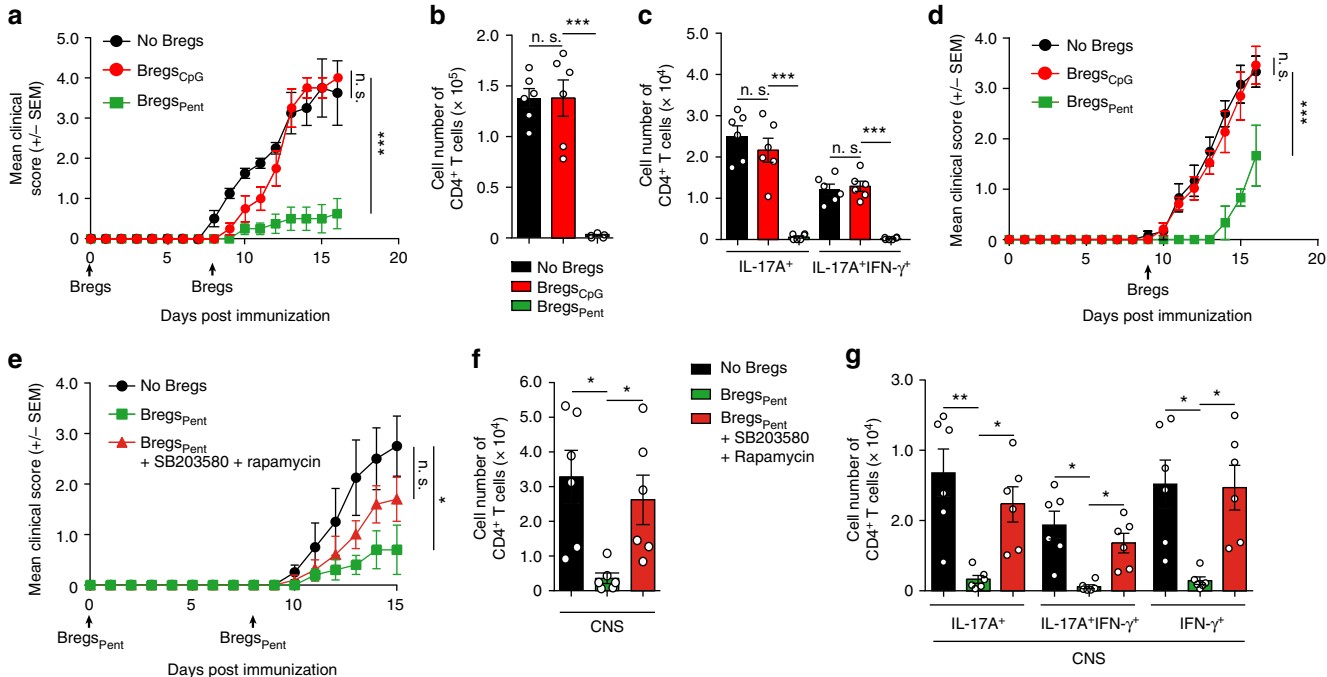

**Fig. 6** Pentanoate-treated Bregs protect mice from EAE development. **a–g** EAE development in mice in the presence or absence of pentanoate-induced Bregs. For induction of EAE, *FIR × tiger* mice were immunized with MOG peptide. A control group of Bregs (Bregs_{CpG}) was treated with vehicle (water). Pentanoate-treated Bregs (Bregs_{Pent}) were generated in the absence or presence of rapamycin (100 nM) and SB203580 (5 μM). Mice received $2 \times 10^6$ sorted GFP+ Bregs on day 0 and day 8 (**a–c** and **e–g**) or only on day 8 (**d**) post immunization ($n = 6$ mice per group from one representative experiment of two). Mice were monitored daily for EAE progression. Total cell number of pathogenic CD4+ T lymphocytes in the CNS on day 17(**a**, **d**) or day 15 (**e**) after EAE development was analyzed by flow cytometry. Black arrows indicate the day of administration of Bregs (**a**, **d**). Error bars indicate SEM, n.s. = not significant; *$P < 0.05$, **$P < 0.01$, ***$P < 0.001$ (one-way ANOVA test)

capacity in Bregs (Fig. 5b). In addition, the expression of hexokinase 2, an enzyme catalyzing the rate-limiting step of glycolysis was enhanced after pentanoate treatment (Fig. 5c). Pentanoate-mediated upregulation of IL-10 expression in Bregs was diminished by treating these cells with 2-DG. Of note, the combination of glycolysis inhibition and blockade of p38 MAPK activity almost completely abolished pentanoate-mediated IL-10 production in Bregs (Supplementary Fig. 5b). Interestingly, only Bregs treated with CpG and pentanoate but not those stimulated with CpG alone enhanced their IL-10 secretion following an increase in glucose supplementation (Supplementary Fig. 5c). Incubation of Bregs with the PDHK inhibitor DCA, which promotes glucose oxidation, increased IL-10 production, while co-treatment of cells with DCA and 2-DG reduced the IL-10 levels (Supplementary Fig. 5d). Similarly to Th17 cells, the treatment of Bregs with pentanoate led to increased levels of phosphorylated AKT and mTOR as well as of downstream targets of mTOR complex 1 such as S6K and S6 (Fig. 5d). The blockade of signaling molecules upstream of mTOR such as PI3K (wortmannin), PDK1 (GSK2334470), and AKT (AKT inhibitor IV) by specific inhibitors led to significant reduction in IL-10 production by pentanoate-treated Bregs (Fig. 5e, f). The combined inhibition of mTOR (rapamycin) and p38 (SB203580) resulted in almost complete blockade of IL-10 production and reduced cell size (glucose metabolism was shown to enhance the cell size in mammals and yeast[39,40]) of pentanoate-induced Bregs (Fig. 5e, f and Supplementary Fig. 6a-c). Normally, the transfer of $2 \times 10^6$ LPS- or CpG-induced Bregs does not suppress the onset of EAE in mice, probably due to their high pro-apoptotic capacity (Supplementary Fig. 4c, d). To analyze if pentanoate-generated Bregs are able to inhibit autoimmune responses in vivo, we induced EAE in WT mice and transferred $2 \times 10^6$ Bregs after

pentanoate treatment on days 0 and 8 (prevention setting) or only on day 8 (therapeutic setting) after immunization. While the untreated group of mice (without Bregs) as well as a cohort of animals that received Bregs treated with vehicle exhibited strong EAE symptoms with a paralysis of limbs and infiltration of immune cells into CNS, mice which received pentanoate-treated Bregs did not develop clinical symptoms of autoimmune disease due to lack of infiltrating, pathogenic lymphocytes in the CNS (Fig. 6a–d). To test the involvement of mTOR complex and p38 MAPK on the suppressive capacity of pentanoate-treated Bregs, we subsequently transferred Bregs generated in the presence of rapamycin and SB203580 into mice, in which EAE was induced with MOG peptide. While Bregs expanded with pentanoate suppressed the autoimmune disease in the CNS, the pretreatment of these cells with mTOR/p38 MAPK inhibitors resulted in markedly reduced ability to protect mice from EAE (Fig. 6e–g). These findings reveal an important role for SCFA pentanoate in improving the suppressive activity of IL-10-secreting Bregs, which might be an interesting therapeutic approach in the future.

## Discussion
In the experimental studies described herein, we have examined whether the SCFA pentanoate is capable of modulating the metabolic pathways in lymphocytes. We here show that this SCFA is present in the gut lumen of conventional but not of GF mice. Recent reports clearly indicate that small molecules produced by commensal bacteria are able to induce metabolic and epigenetic reprogramming linked to suppression of inflammatory immune responses[6,16]. Acetate, propionate, and butyrate are the most abundant SCFAs generated in the intestinal lumen by fermentative activity of anaerobic commensals[1]. In particular,

several functional studies provided a strong evidence for important role of butyrate in the induction of colonic Tregs[3,5,6].

The main aim of this study was to explore possible therapeutic effects of pentanoate on inflammatory and autoimmune disorders. Notably, we found that butyrate exhibit profound toxic effects on lymphocytes by strongly inducing apoptosis in Bregs. In contrast to treatment of Bregs with pentanoate, we were not able to generate and stabilize suppressive Breg cell phenotype by using butyrate. Thus, although beneficial effects of dietary fiber-dependent SCFAs on chronic inflammatory diseases are well known, we here show that pentanoate might be even more suitable for inducing anti-inflammatory responses since it is less toxic than butyrate and much more potent than acetate. Of note, in mammals, the architecture of the colonic crypts has been shown to provide a metabolic barrier against high levels of luminal butyrate. Differentiated colonocytes located at the top of mammalian crypts consume butyrate as an energy source and protect stem and progenitor cells at the base of the crypt from butyrate-mediated toxic effects[41].

A better understanding of the gut microbiota-derived signals that cause metabolic and epigenetic changes in immune cells will encourage therapeutic strategies aimed at developing novel anti-inflammatory agents. Our study reveals previously unknown metabolic rewiring caused by the impact of pentanoate on B cells and CD4[+] effector T lymphocytes, leading to increased glycolysis, pyruvate oxidation, and secretion of immunomodulatory cytokine IL-10. Intriguingly, pentanoate was not able to potentiate the expansion of mucosal Tregs in spite of its strong HDAC-inhibitory activity. The anti-inflammatory effect of pentanoate was not only reflected in strong upregulation of IL-10 production in Bregs and CD4[+] effector T cells but also by a potent suppression of pathogenic Th17 cell phenotype. Normally, upon activation of T cells, the metabolic reprogramming towards glycolysis supports their inflammatory phenotype and production of cytokines such as IFN-γ or IL-17A[42,43]. Remarkably, the treatment of Th17 cells with pentanoate increased already active glycolytic pathway, thereby influencing the immunological fate of these cells by strongly inducing IL-10 production. This finding might be important to understand how fine-tuning of the cellular metabolic pathways influences the function of effector T cells. By using specific inhibitors for various metabolically active enzymes, we found that pentanoate-dependent secretion of IL-10 was dependent on enhanced mTOR activity, glucose-derived pyruvate oxidation and increased acetyl-CoA production. Recently, the metabolic tracing analysis of memory CD8[+] T cells treated with acetate has shown that this SCFA was able to expand the acetyl-CoA pool in lymphocytes by acting as a direct substrate for acetyl-CoA production and histone acetylation. In these experimental settings, the inhibition of ACLY (enzyme catalyzing the conversion of citrate to acetyl-CoA) led to a decrease in IFN-γ expression and reduced memory T cell function[44]. In Th17 cells, we observed two distinct effects of pentanoate on the metabolic and epigenetic state of lymphocytes that together improved the anti-inflammatory capacity of these cells. On the one hand, the increased glycolytic rate and pyruvate oxidation induced IL-10 production and the blockade of ACLY led to the decrease of IL-10 secretion. On the other hand, the HDAC-inhibitory activity of pentanoate predominantly suppressed the IL-17A expression. Interestingly, two novel studies have shown that butyrate is able to ameliorate rheumatoid arthritis and colitis by targeting HDAC8 and HDAC1 in Th17 cells, respectively[45,46]. It is possible that the structurally similar molecule, pentanoate, also inhibits class I HDACs such as HDAC8 and HDAC1, which might lead to the downregulation of Il17a expression.

In summary, these data demonstrate that the SCFA pentanoate exerts its immunomodulatory effects by metabolic and epigenetic reprogramming of T and B lymphocytes. As a consequence of the metabolic and epigenetic modulation of the balance between pro- and anti-inflammatory immune responses, pentanoate was capable of ameliorating T cell-mediated immunopathology in the gut and brain of mice. These findings suggest that the gut microbiota-derived metabolite pentanoate might be deployed therapeutically as a low-cost and well-tolerated substance in patients with multiple sclerosis (MS) and other Th17-mediated autoimmune disorders. Future studies will be needed for evaluating therapeutic potential of pentanoate in humans.

## Methods

**Mice**. 8–12-week-old WT mice (C57BL/6 background) were kept under specific pathogen-free (SPF) or germ-free (GF) conditions. GF animals were reared in sterile plastic isolators and were monitored biweekly for sterility. FIR × tiger reporter mice as well as Rag1[−/−] animals on C57BL/6 background were bred at the animal facility of the Biomedical Research Center, University of Marburg, Germany. All experiments were performed in accordance with the relevant regulatory standards. Animal work was approved by Regierungspräsidium Gießen, Germany under project numbers 70/2014 and EX7-2015.

**In vivo application and measurement of SCFAs**. Oral treatment of FIR × tiger and GF mice with SCFAs was performed according to a published protocol[8] by providing sodium butyrate or sodium pentanoate into the drinking water (both 150 mM) to GF and FIR × tiger mice for 4 weeks. After 4 weeks of treatment, lamina propria mononuclear cells (LPMCs) were isolated using collagenase digestion (0.4 mg/ml collagenase D and collagenase VIII, both Sigma-Aldrich) and 40/70% Percoll density gradient. The analysis of colonic T lymphocytes was performed by flow cytometry. Approximately 20 mg of fecal content derived from the small intestine, cecum, and colon of WT and GF mice were homogenized and centrifuged for 10 min at 21,000g. The obtained supernatants were derivatized by using AMP+ Mass Spectrometry Kit (Cayman Chemical) and diluted with milliQH$_2$O (1:10). Subsequently, samples were separated by applying a reversed-phase column (C8: 1.7 µm, 2.1 × 150 mm, Acquity™ UPLC BEH™, Waters). The analysis was performed by ultra-high performance liquid chromatography (UHPLC) coupled to mass spectrometry (MS)[19]. The measurement protocol for gas chromatography-mass spectrometric (GC-MS) analysis of SCFAs was published by us[47].

**In vitro generation and adoptive transfer of Bregs**. Splenic B cells were enriched by depletion of non-B cells and cultured in RPMI medium containing 4 µg/ml CpG 2395 (TIB MOLBIOL) or 5 µg/ml LPS (Sigma-Aldrich) for 4 days. On day 1 of the cell culture, B cells were either treated with 5 mM pentanoate or left untreated for the next 3 days. Cells were routinely tested for the expression of CD19 and B220 by FACS analysis. In some experiments, Bregs were treated with p38 MAPK inhibitor SB203580 (1–10 µM, Enzo Life Sciences), 2-DG (1 mM, Sigma-Aldrich), or with following metabolic inhibitors: rapamycin (100 nM, Sigma-Aldrich), wortmannin (350 µM, Sigma-Aldrich), GSK2334470 (200 nM, Tocris Bioscience) and AKT inhibitor IV (75 nM, EMD Millipore). To monitor glucose supplementation-induced changes in Bregs, cells were supplemented with various glucose concentrations (1–5 mM, Carl Roth). For induction of T cell-mediated colitis, at day 0, naive CD4[+] T cells were isolated from spleens and LNs of WT mice and transferred i.p. into Rag1[−/−] mice (0.5 × 10⁶ naive CD4[+] T cells/mouse). In the second cohort of Rag1[−/−] immunodeficient recipients, naive T cells were co-transferred at a ratio 1:4 together with in vitro polarized Bregs pre-cultured for 4 days with 5 mM sodium pentanoate and 4 µg/ml CpG. Throughout the experiment, mice were monitored for clinical signs of intestinal inflammation. On day 50 after adoptive transfer of lymphocytes, T cells were purified from lamina propria and mLN of Rag1[−/−] mice and analyzed by FACS analysis.

**ATP assay**. Bregs were induced with 4 µg/ml CpG in the presence or absence of 5 mM pentanoate. After 4 days, Bregs were harvested and pelleted after adjustment to 1 × 10⁶ per assay. The pellets were analyzed using a bioluminescent ATP assay kit (Sigma-Aldrich) according to the manufacturer's protocol.

**Acetyl-CoA assay**. Splenic CD4[+] T cells were isolated from WT mice and polarized towards Th17 cells. T cells were treated with 5 mM pentanoate 24 h post activation and cultured for 3 days. After harvesting, the cells were adjusted to 10 × 10⁶ per assay. The pellets were analyzed using a fluorometric acetyl-CoA assay kit (Sigma-Aldrich) according to the manufacturer's protocol.

**In vitro generation of Th17 cells and Tregs**. CD4[+] T lymphocytes were purified from the murine spleens and LNs using the kit for negative isolation of T cells. Th17 cells were generated with plate-bound anti-CD3 (5 µg/ml, 145-2C11) and soluble anti-CD28 (1 µg/ml, 37.51) in the presence of 5 µg/ml anti-IFN-γ (XMG1-2), anti-IL-4 (10% culture supernatant of clone 11B11), 25 ng/ml IL-6 and 1 ng/ml TGF-β1 (both Peprotech). When indicated, Th17 were stimulated with sodium

butyrate (1 mM), sodium pentanoate (0.5–5 mM) or 1-10 nM TSA (Sigma-Aldrich). Some Th17 cells were additionally treated with 100 nM rapamycin, 1 μM 2-DG, or with following metabolic inhibitors: shikonin (100 nM, Sigma-Aldrich), SB204990 (5 μM, Tocris Bioscience), DCA (7.5 mM, Sigma-Aldrich), metformin (2.5 mM, Sigma-Aldrich) and AKT inhibitor IV (75 nM). For Treg generation, purified CD4$^+$ T cells were primed with plate-bound anti-CD3 (5 μg/ml) and soluble anti-CD28 (1 μg/ml) in combination with 5 μg/ml anti-IFN-γ, 10% culture supernatant of anti-IL-4, 2 ng/ml TGF-β1, and 50 U/ml IL-2.

**Flow cytometry.** For FACS analysis, single-cell suspensions were stained with the following antibodies: anti-CD4 (BioLegend, clone GK1.5, #100402, 1:400), anti-CD4 (BD Biosciences, clone RM4-5, #560468, 1:500), anti-CD19 (Thermo Fisher Scientific, clone eBio1D3, #45-0193-82, 1:400), and anti-CD8 (BD Biosciences, clone 53-6.7, #560776, 1:500). For intracellular staining, T cells were restimulated with 50 ng/ml PMA and 750 ng/ml ionomycin in the presence of 10 μg/ml brefeldin A (all from Sigma-Aldrich) for 4 h. After fixation and permeabilization, the cells were stained with anti-IL-17A (Thermo Fisher Scientific, clone eBio17B7, #17-7177-81, 1:200) and anti-IFN-γ (Thermo Fisher Scientific, clone XMG1.2, #17-7311-82, 1:500). For detection of apoptotic cells, the cultured Bregs were harvested, washed with HBSS buffer and resuspended in a HBSS solution containing FITC-labeled Annexin V (Thermo Fisher Scientific, #88-8005-72, 1:100). To measure the cell proliferation, the staining with dye carboxyfluorescein succinimidyl ester (5 μM CFSE, Sigma-Aldrich) was performed. The cells were analyzed using FACSCalibur cytometer or BD FACSAria III cell sorter (both BD Biosciences). Data were analyzed with FlowJo analysis software (TreeStar). All FACS sequential gating/sorting strategies for the analysis of T and B cells are provided in Supplementary Fig. 7.

**Mouse EAE model.** For EAE induction, *FIR × tiger* mice were subcutaneously (s.c.) injected in the abdomen with 200 μg MOG$_{37-50}$ peptide (synthesized by R. Volkmer, Charité, Berlin, Germany) emulsified in CFA (containing 500 μg *Mycobacterium tuberculosis* H37RA, Difco Laboratories). At the day 0 of immunization and 2 days later, 200 ng pertussis toxin (Sigma-Aldrich) was intraperitoneally (i.p.) administered. To analyze the impact of pentanoate on the development of EAE, animals were daily i.p. treated with 800 mg sodium pentanoate/kg mice throughout the duration of the experiment. For adoptive transfer of Bregs, the EAE was induced by immunizing mice as described above. On days 0 and 8 after EAE induction, $2 × 10^6$ of IL-10$^+$ CpG-induced Bregs, generated in the presence of pentanoate alone or pentanoate + 5 μM SB203580 + 100 ng rapamycin were injected i.p. into the recipient mice. After 15–20 days, the animals were analyzed for the cells infiltrating brain and spinal cord by FACS analysis. In some experiment, EAE was induced in GF mice monocolonized with SFB by applying the protocol described above. The intestinal colonization of GF animals with SFB was routinely tested. One group of monocolonized GF mice was treated with 800 mg sodium pentanoate/kg mice (three times per week, i.p. injection). The disease severity was scored using a published scoring system[48].

**Fluorescence microscopy.** Immunofluorescence imaging was performed by using the scanning confocal microscope Leica SP8i equipped with ×63 (NA 1.4) and ×100 (NA 1.35) oil immersion objectives. Th17 cells and Bregs derived from *FIR × tiger* mice were generated in the presence or absence of pentanoate as described. Th17 cells were restimulated prior to fixation with 2% methanol-free formaldehyde in PBS and stained with CD4-V450 (BD Bioscience) and IL-17A-APC (eBioscience). The cytoplasm of Bregs was stained using CellTracker Blue CMAC (Thermo Fisher) prior to analysis. Analysis and quantification were performed using Leica Software LAS X and ImageJ software (NIH) by calculating the corrected total cell fluorescence (CTCF). The cell area (μm$^2$) for Th17 cells and Bregs was determined using ImageJ software. 100 cells per group were analyzed. The experiment was repeated three times.

**ChIP analysis.** Pathogenic Th17 cells were generated from purified CD4$^+$ T lymphocytes by stimulating them with 25 ng/ml IL-6, 25 ng/ml IL-23, and 20 ng/ml IL-1β (in addition to anti-CD3/CD28, anti-IFN-γ and anti-IL-4 Abs). On day 2 of the cell culture, fixation was performed with 1% methanol-free formaldehyde in PBS for 10 min at room temperature followed by quenching with 125 mM glycin for 5 min. Cells were washed with ice-cold PBS twice and harvested using a cell scraper. The pellet was lysed in hypotonic buffer L1 (5 mM PIPES pH 8.0), 85 mM KCl, 0.5% (v/v) NP40, protease inhibitor mix (Sigma, no. P8340, 1:1000) for 20–40 min on ice. Nuclei were resuspended in ChIP RIPA buffer (10 mM Tris–HCl pH 7.5, 150 mM NaCl, 1% NP40 (v/v), 1 mM EDTA) supplemented with 1:1000 protease inhibitor mix (Sigma), incubated on ice for 10–20 min and sheared with a Branson S250D Sonifier (Branson Ultrasonics) using a microtip in 1 ml aliquots in 15 ml conical tubes. 50 pulses of 1 s, 4 s pause, 20% amplitude were applied with cooling of the sample in a 15 ml tube cooler (Active Motif, no. 53077). A 15 min 20,000×g supernatant was precleared with 10 μg of IgG coupled to 100 μl of blocked sepharose slurry per milliliter of sheared chromatin (see below) for 45 min at 4 °C with agitation. IPs were carried out with precleared chromatin equivalent to $6 × 10^6$ cells. For precipitation, protein A sepharose (GE Healthcare Life Sciences, no. 1752800) was washed twice with ChIP RIPA buffer and blocked with 1 g/l BSA and 0.4 g/l sonicated salmon sperm DNA (Life Technologies, no. 15632011)

overnight. 50 μl of blocked bead slurry (1:1 volume ratio with liquid phase) were used per IP. Samples were washed once in buffer I (20 mM Tris pH 8.1; 150 mM NaCl; 1% (v/v) Triton X-100; 0.1% (v/v) SDS; 2 mM EDTA), once in buffer II (20 mM Tris pH 8.1; 500 mM NaCl; 1% (v/v) Triton X-100; 0.1% (v/v) SDS; 2 mM EDTA), twice in buffer III (10 mM Tris pH 8.1; 250 mM LiCl; 1% (v/v) NP40; 1% (w/v) sodium deoxycholate; 1 mM EDTA) on ice and twice in Qiagen buffer EB (no. 19086) at room temperature. Immune complexes were eluted twice with 100 mM NaHCO$_3$ and 1% SDS (w/v) under agitation. Eluates were incubated overnight at 65 °C after adding 10 μg of RNase A and 20 μg of proteinase K in the presence of 180 mM NaCl, 35 mM Tris–HCl pH 6.8 and 9 mM EDTA. Input samples representing 1% of the chromatin used per IP were reverted in parallel. Samples were purified using the Qiagen PCR purification kit according to the manufacturer's instructions. Quantitative PCR was performed in three technical replicates per sample with the ABsolute SYBR Green master mix (Thermo Scientific, no. AB-1158B) in in Mx3000p and Mx3005 thermocyclers (Agilent) using the following primers: *Il10* promoter fw AGGGAGGAGGAGGAGCCTGAATA, rv ATGGAGCTCTCTTTTCTGCAAG.

**HDAC activity assay.** CD4$^+$ T cells were isolated from spleen and LNs of WT mice. Afterwards, cell lysates were prepared and incubated with 5 mM of indicated SCFAs or 0.5 μM TSA for 1 h at 37 °C to allow for HDAC inhibition. Subsequently, the peptide substrate Ac-Arg-Gly-Lys(Ac)-AMC (Bachem) was added to the lysates for 30 min at 37 °C followed by addition of stop solution containing trypsin (30 min, 37 °C). HDAC activity was measured at the spectrofluorometer FLUOstar Omega (BMG Labtech).

**Western blot analysis.** Cell lysates were obtained from in vitro generated lymphocytes and SDS-PAGE was performed. After the transfer to a PVDF membrane (Millipore), the samples were incubated with Abs. For the assessment of AKT and mTOR activation, Bregs and Th17 cells treated with pentanoate were analyzed with following Abs: anti-phospho-AKT$^{Ser437}$ (#9271, Cell Signaling Technology, 1:1000), anti-phospho-AKT$^{Thr308}$ (#13038, Cell Signaling Technology, 1:1000), anti-total AKT (#9272, Cell Signaling Technology, 1:1000), anti-phospho-mTOR$^{Ser2448}$ (#5536, Cell Signaling Technology, 1:1000), anti-total mTOR (#2983, Cell Signaling Technology, 1:1000), anti-phosphoS6K$^{T389}$ (#9205, Cell Signaling Technology, 1:1000), anti-total S6K (#9202, Cell Signaling Technology, 1:1000), anti-phosphoS6$^{S235/236}$ (#2211, Cell Signaling Technology, 1:1000), and anti-total S6 (#2217, Cell Signaling Technology, 1:1000). The original uncropped scans of all western blots are shown in Supplementary Fig. 8.

**Measurement of ECAR.** Bregs generated with CpG and T cells differentiated into Th17 cells or Tregs were cultured in the presence or absence of 5 mM pentanoate. ECAR was analyzed with the XF96 Analyzer (Seahorse Biosciences). $2 × 10^5$ lymphocytes/well were used for ECAR measurement. Basal ECAR reading was performed with cells cultured in base DMEM without the addition of glucose. ECAR was measured with final concentrations of 10 mM glucose, 2 μM oligomycin (Seahorse Biosciences), and 100 mM 2-DG. To determine the instrumental background, separate control wells without biological samples were used. For the measurement of glycolytic capacity, the difference between ECAR values following oligomycin injection and baseline ECAR reading was used.

**RNA-seq analysis.** RNA was purified from in vitro generated Bregs or Th17 cells using the RNeasy Mini kit (Qiagen). The purified RNAs were sequenced on an Illumina HiSeq 1500 device. Reads were aligned to the *Mus musculus* genome retrieved from Ensembl revision 83 (mm10) with STAR 2.4. Tag counts were calculated and normalized to one million mapped exonic reads and gene length (FPKM). To generate the set of expressed genes, only genes with a minimum read count of 50 and a minimum FPKM of 0.3 were kept. DEseq2 (version 1.12.3) was used to determine differentially expressed genes. Only genes with an increase or decrease of at least two-fold according to DESeq2-analysis and with a maximum FDR of 0.05 were considered as regulated genes. Gene set enrichment analysis (GSEA) was performed using the KEGG Pathway Database. Sequencing data were deposited at EBI ArrayExpress under accession numbers E-MTAB-6114 for Th17 cells and E-MTAB-6115 for Bregs. The transcriptional signature of murine Th17 cells was published[49].

**ELISA.** The secreted cytokines were measured in the culture supernatants using standard protocols. Murine ELISA (purchased from BD Biosciences) were used for detection of IL-10 secretion.

**qRT-PCR.** Following extraction of total RNA from the investigated cells using the High Pure RNA Isolation Kit (Roche), RNA was transcribed into cDNA with RevertAid First Strand cDNA Synthesis Kit (Thermo Scientific). Quantitative real-time PCR was performed at the StepOnePlus machine (Applied Biosystems) with following primers: *HK2* fw TGATCGCCTGCTTATTCACGG, *HK2* rv AACCGCC TAGAAATCTCCAGA, *Il10* fw ACAACATACTGCTAACCGACTCC, *Il10* rv CA AATGCTCCTTGATTTCTGGGC. The expression levels of examined genes were normalized to the expression of *Hprt1*. The following primer set was used: *Hprt1* fw

CTGGTGAAAAGGACCTCTCG, *Hprt1* rv TGAAGTACTCATTATAGTCAAG
GGCA.

**Statistical analysis**. Except for RNA-seq experiments, all statistical analyses were performed using GraphPad Prism 5 program (GraphPad). The statistical significance of the results was evaluated by the one-way analysis of variance (ANOVA) test (for the comparison of the multiple groups) and Student's *t*-test. Data are represented as the mean ± SEM with following *P*-values: ***$P < 0.001$, **$P = 0.001–0.01$, *$P = 0.01–0.05$.

**Reporting summary**. Further information on experimental design is available in the Nature Research Reporting Summary linked to this article.

## Data availability

RNA-seq data have been deposited in ArrayExpress Archive of Functional Genomics Data under the accession codes E-MTAB-6114 for Th17 cells and E-MTAB-6115 for Bregs. All other relevant data are available from authors upon reasonable request.

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

## Acknowledgements

The authors are grateful to Dr. Alesia Walker (Helmholtz Zentrum München, Neuherberg, Germany) for SCFA analysis. The authors thank Elena Jenike and Anne Hellhund for technical assistance. The authors also thank Dr. Wolfgang Meißner and Dr. Julia Obert for establishing the measurement of ECAR as well as Dr. Katrin Roth and members of the Core Facility Microscopy, University of Marburg for providing microscopic devices and for excellent technical support. This study was supported

by a research grant from the Fritz Thyssen Foundation (Alexander Visekruna) and by Studienstiftung des deutschen Volkes and Von Behring-Röntgen-Stiftung (Maik Luu).

## Author contributions

A.V. and U.S. conceived the study. M. Luu and A.V. wrote the article. M. Luu, S.P., V.K., N.V., and L.C.C. designed and performed in vitro experiments and analyzed the data. M. Luu, R.R., and N.V. performed in vivo experiments. H.R., B.L., A.N., and T.S. were involved in the RNA Seq analyses. R.S. and M. Lauth performed experiments dealing with the activation of mTOR. S.L., J.H., and M.M.Z. performed and interpreted GC-MS analysis. T.A. performed ChIP analysis. Y.S., M. Luu, and B.L. performed bioinformatic analyses.

## Additional information

**Competing interests:** The authors declare no competing interests.

