## [Peer Review File · Nature Communications]

Reviewers' comments:

Reviewer #1 (Short-chain fatty acid, IBD, gut immunity; [redacted])(Remarks to the Author):

The authors addressed a part of the concerns raised by the reviewer in the revised manuscript. However, several amendments are still necessary before I recommend the manuscript for publication.

Major issues

Pentanoate is a commensal-derived metabolite, and its concentration in the lumen is less than 1.0 mM (Sup. Fig. 1a). These data suggest that pentanoate levels in the gut tissues and systemic are estimated to be far below the luminal concentration under the physiological conditions. Because the level of acetate in serum after the administration of 150 mM of acetate in drinking water is at 0.1 mM (Lucas S et al., Nat Commun. 2018), administration of 150 mM pentanoate in drinking water also may not dramatically increase the systemic concentration of pentanoate. In the EAE experiments presented here, 800 mg/Kg BW of pentanoate was intraperitoneally injected to mice. In this case, the pentanoate level could reach around 5 mM (the default concentration for in vitro experiment in this study), which should impact on function and differentiation of T and B cells. Therefore, the aim of this study deems to evaluate the therapeutic effect of pentanoate rather than assess the physiological role of this SCFA. The authors should clearly describe this point in the manuscript.

In Figure 2, the author mentioned that pentanoate promotes IL-10 (GFP) production in Th17 cells by enhancing glucose oxidation and supplying acetyl-CoA to increase acetylation of H4 histone at the II10 promoter region. To conclude this, please show II10 gene expression in Fig. 1b (RNA-seq heat map).

Pentanoate increases acetylation of H4 histone at the II10 promoter region. Is the accumulation of H4 histone acetylation specific to the II10 promoter region? What about other genes including Th17-related II17a, Rorc and so on.

Pentanoate reduces II17a expression by inhibiting HDACs (Fig. 2a,b). The previous study has reported that HDAC8 inhibition is essential to downregulate II17a expression (Kim DS et al., Front Immunol. 2018). Does pentanoate inhibit HDAC8 to downregulate II17a expression? Please at least discuss this possibility.

TSA (a pan HDAC inhibitor) suppressed the expression of II17a but not II10 in Th17 differentiation system at a concentration of 500 nM (Fig. 2a,b). In Fig. 5c, authors show that pentanoate increases IL-10 production in Breg cells, but TSA has no effect on II10 production in Breg cells when treated at the very lower concentration (5 or 10 nM). Based on these data, the author concludes that HDACi activity of pentanoate is not responsible for the promotion of II10 expression. However, it might be still early to consider that TSA does not affect II10 expression in Breg cells unless treated with a higher concentration (i.e., 500 nM). Butyrate is known to enhance II10 expression by inhibiting HDAC1 (Liao HY et al., Sci Rep. 2016), suggesting TSA and pentanoate possibly induces II10 expression. The authors should carefully address whether the HDACi activity of pentanoate is related to Breg II10 production or not.

In Fig. 5, Pentanoate is shown to promote II10 expression of Breg cells by enhancing glycolysis. Breg cells are known to abundantly produce IL-10, which is critical for preventing EAE

development. However, it is unclear pentanoate-induced IL-10 production is essential to ameliorate the disease (Fig. 6). To prove this, Breg cells treated with vehicle (water or sodium chloride?) should be employed as a control of the EAE experiments in Fig. 6a-d. Only vehicle injection without Breg cells is not appropriate as the control. Additionally, if possible, butyrate, acetate, or TSA-treated Breg cells might give authors further information.

In the EAE experiment, upregulation of IL-10 in pentanoate-pretreated Breg cells is maintained in the body after the cell transfer?

Minor issues

Acetylated Arg (Ac-Arg-Gly-Lys-AMC) was used as a substrate for HDAC activity assay. Are there any particular reasons not to use Ac-Lys?

What ingredient is the substrate for pentanoate fermentation?

In Fig. 2d (and others, too), the signature Pentanoate, 2-DG on the upper region of the graph should be located below the graph.

Reviewer #3 (Th17, IBD; [redacted])(Remarks to the Author):

Authors responded to my comments faithfully. Most of the concerns were adequately answered.

Referee 1 ([redacted])

The authors addressed a part of the concerns raised by the reviewer in the revised manuscript. However, several amendments are still necessary before I recommend the manuscript for publication.

Major issues

REFeree 1

Pentanoate is a commensal-derived metabolite, and its concentration in the lumen is less than 1.0 mM (Sup. Fig. 1a). These data suggest that pentanoate levels in the gut tissues and systemic are estimated to be far below the luminal concentration under the physiological conditions. Because the level of acetate in serum after the administration of 150 mM of acetate in drinking water is at 0.1 mM (Lucas S et al., Nat Commun. 2018), administration of 150 mM pentanoate in drinking water also may not dramatically increase the systemic concentration of pentanoate. In the EAE experiments presented here, 800 mg/Kg BW of pentanoate was intraperitoneally injected to mice. In this case, the pentanoate level could reach around 5 mM (the default concentration for in vitro experiment in this study), which should impact on function and differentiation of T and B cells. Therefore, the aim of this study deems to evaluate the therapeutic effect of pentanoate rather than assess the physiological role of this SCFA. The authors should clearly describe this point in the manuscript.

AUTHOR RESPONSE

We thank the reviewer for raising this very important issue and for giving us the opportunity to highlight translational implications in the study. We are grateful for all constructive suggestions. We are aware that physiological role of pentanoate is not the main focus of our study. In light of our novel observations and considering a robust therapeutic potential of pentanoate, it is likely that this molecule should be considered as a candidate for future therapies in autoimmune diseases.

According to the reviewer's comment, we have now clearly stated that the aim of this study was to investigate the therapeutic effect of pentanoate. I have included two following sentences in the main text: "The aim of this study was to evaluate the capacity of the SCFA pentanoate (valerate, C5) as a potential therapeutic for autoimmune and inflammatory diseases." (Introduction) and "The main aim of this study was to explore possible therapeutic effects of pentanoate on inflammatory and autoimmune disorders." (Discussion). This is further clarified in the last sentence of the abstract.

REFeree 1

In Figure 2, the author mentioned that pentanoate promotes IL-10 (GFP) production in Th17 cells by enhancing glucose oxidation and supplying acetyl-CoA to increase acetylation of H4

histone at the *Il10* promoter region. To conclude this, please show *Il10* gene expression in Fig. 1b (RNA-seq heat map).

AUTHOR RESPONSE

As suggested by the reviewer, we have included *Il10* gene expression into the RNA-seq heat map (please see new Figure 1b).

REFeree 1

Pentanoate increases acetylation of H4 histone at the *Il10* promoter region. Is the accumulation of H4 histone acetylation specific to the *Il10* promoter region? What about other genes including Th17-related *Il17a*, *Rorc* and so on.

AUTHOR RESPONSE

We performed additional experiments as suggested by the reviewer and did not find an increased acetylation of H4 at the promoter region of Th17-associated genes such as *Il17a*, *Il17f*, *Rorc* and *Rora*. In addition, we also analyzed some genes that are not related to Th17 cells and we obtained similar results (e.g. *Ccnd2* and *Gramd1b* genes, data not shown). These data are now presented in the attached Fig. 1 for the Reviewer. We cannot exclude the possibility that other genes are affected by pentanoate treatment but the increase in acetylation of H4 histone at the *Il10* promoter region seems to be relatively specific.

Fig 1. The acetyl-H4 levels at the promoter region of Th17-associated genes *Il17a*, *Il17f*, *Rorc* and *Rora* were determined by ChIP assay. Error bars indicate SEM; n. s. = not significant (Student's *t*-test). Three similar experiments were performed. The analysis was performed with the same protocol as described for *Il10* promoter region (please see the part Methods in the revised manuscript).

The following primer sequences were used for the ChIP experiments:

Il17a fw CACCTCACACGAGGCACAAG
Il17a rw ATGTTTGCGCGTCCTGATC

Il17f fw GTTCTCCAATGGCTGCTTC
Il17f rv AGTGGAAACAGGGACAGTGATT
Rorc fw CAGAAACACTGGGGGAGAGC
Rorc rv ACACAGCTGGCAGTGGAGG
Rora fw GCAAGGCAGAGAGCTTCCG
Rora rv CACCAAAGTCCCTCGCCAC

REFeree 1

Pentanoate reduces *Il17a* expression by inhibiting HDACs (Fig. 2a,b). The previous study has reported that HDAC8 inhibition is essential to downregulate *Il17a* expression (Kim DS et al., Front Immunol. 2018). Does pentanoate inhibit HDAC8 to downregulate *Il17a* expression? Please at least discuss this possibility.

AUTHOR RESPONSE

We thank the reviewer for this useful comment and agree that predominantly the inhibition of class I HDACs such as HDAC8 (Kim D.S. *et al.*, Front. Immunology, 2018) and HDAC1 (as recently suggested by Zhou L. *et al.* Inflamm. Bowel Dis., 2018) might be involved in the observed effect of pentanoate. We have discussed this possible mechanism in the discussion part. We also cited the work of Kim D.S. *et al.* as well as that of Zhou L. *et al.*

REFeree 1

TSA (a pan HDAC inhibitor) suppressed the expression of *Il17a* but not *Il10* in Th17 differentiation system at a concentration of 500 nM (Fig. 2a,b). In Fig. 5c, authors show that pentanoate increases IL-10 production in Breg cells, but TSA has no effect on *Il10* production in Breg cells when treated at the very lower concentration (5 or 10 nM). Based on these data, the author concludes that HDACi activity of pentanoate is not responsible for the promotion of *Il10* expression. However, it might be still early to consider that TSA does not affect *Il10* expression in Breg cells unless treated with a higher concentration (i.e., 500 nM). Butyrate is known to enhance *Il10* expression by inhibiting HDAC1 (Liao HY et al., Sci Rep. 2016), suggesting TSA and pentanoate possibly induces *Il10* expression. The authors should carefully address whether the HDACi activity of pentanoate is related to Breg *Il10* production or not.

AUTHOR RESPONSE

We appreciate the comment of the reviewer and have duly addressed the concern by repeating many of the key experiments. Only in Fig. 2a, in a biochemical assay with cell lysates (but not with alive cells) we have used 500 nM TSA. In the Fig. 2b, we used 10 nM TSA. Both concentrations (500 nM TSA for Fig. 2a for a biochemical assay as well as 10 nM TSA for Fig. 2b) are indicated in the original Figure legend (Figure legend for Fig. 2 of the manuscript).

The reason why we worked with max. 10 nM TSA is shown in the Fig. 2 for the Reviewer. 500 nM TSA is highly toxic for cells. TSA has a strong pro-apoptotic activity in both B and T lymphocytes at the concentration of 30 nM. At the concentration of 50 nM, nearly 100 % of TSA-treated Bregs are dead. Even very robust cells such as HeLa cervical cancer cells are highly prone to the apoptosis after TSA treatment (by using 100 nM TSA, 90 % of all HeLa cells undergo apoptosis, Bo Ra You and Woo Hyun Park, International Journal of Oncology, 2013).

We carefully re-analyzed concentration-dependent effects of three different HDAC inhibitors (TSA, valproate and SAHA) on Bregs (IL-10 expression and apoptotic rate). In contrast to SCFAs, they are not able to act as a precursor molecule for cellular acetyl-CoA. None of these synthetic HDAC inhibitors was able to strongly induce IL-10 at concentrations that do not

induce strong apoptosis (please see the Fig. 2a-d for the Reviewer). We have repeated these experiments several times with more animals and the tendency is that the level of IL-10 increases very slightly at concentrations that were not toxic for TSA and SAHA, but this trend is only marginal as compared to pentanoate.

In contrast, the effect of pentanoate on IL-10 expression is robust, statistically significant and highly reproducible. At the concentration of 5 mM, pentanoate-treated Bregs and Th17 were quite viable and they strongly increased the production of IL-10. All novel data indicating dose-dependent effects are shown in the Fig. 2 for the Reviewer. Apoptosis-related data are included in this Figure. We cannot exclude a slight effect of TSA on the expression of IL-10. Therefore, in the text we changed the phrase: “TSA was not capable of increasing IL-10 secretion in Bregs” into the following one: “TSA was not capable of strongly increasing IL-10 secretion in Bregs”.

Fig 2. (A) Bar graphs indicate secreted amounts of IL-10 by Bregs incubated for 3 days with three different HDAC inhibitors, TSA, valproate and SAHA. Three independent experiments were performed. (B-D) The percentage of Annexin V⁺ cells was determined by FACS analysis on day 3 in Breg cell cultures treated with indicated concentrations of TSA, valproate and SAHA. Three experiments were performed. Error bars indicate SEM.

REFeree 1

In Fig. 5, Pentanoate is shown to promote IL10 expression of Breg cells by enhancing glycolysis. Breg cells are known to abundantly produce IL-10, which is critical for preventing EAE development. However, it is unclear pentanoate-induced IL-10 production is essential to ameliorate the disease (Fig. 6). To prove this, Breg cells treated with vehicle (water or sodium chloride?) should be employed as a control of the EAE experiments in Fig. 6a-d. Only vehicle injection without Breg cells is not appropriate as the control. Additionally, if possible, butyrate, acetate, or TSA-treated Breg cells might give authors further information.

In the EAE experiment, upregulation of IL-10 in pentanoate-pretreated Breg cells is maintained in the body after the cell transfer?

AUTHOR RESPONSE

We appreciate the suggestion of the reviewer and now provide the control Bregs (Bregs treated with vehicle) in all experiments (in Fig 6a-d). These results were repeated several times and we have never observed that Bregs treated with vehicle (water) had any protective effects on EAE development. We have included these important data in a new Figure 6 (please see novel Fig. 6a-d). We apologize for lack of clarity and for incompleteness.

We thank the reviewer for the suggestion that diverse SCFAs in EAE model and other autoimmune models could possibly be tested. We share this enthusiasm; we hope that there will be significant follow up to this study and that experts in the field will extrapolate these observations to other medically relevant models.

We have routinely checked for the presence of transferred pentanoate-treated Bregs and their IL-10 expression through all EAE experiments. IL-10 production in pentanoate-treated Bregs was maintained in the spleen and draining lymph nodes for at least 15 days after starting EAE protocol.

Minor issues

REFeree 1

Acetylated Arg (Ac-Arg-Gly-Lys-AMC) was used as a substrate for HDAC activity assay. Are there any particular reasons not to use Ac-Lys?

AUTHOR RESPONSE

We thank the reviewer and apologize for this oversight. We have contacted the company (Bachem) and changed the text accordingly. The HDAC tripeptidic substrate is the following one: Ac-Arg-Gly-Lys(Ac)-AMC. It was especially designed for HDAC activity assay and it can also be purchased by Sigma-Aldrich and R&D Systems. The N-acetyl group at the Arg acts as protection group for the tripeptide.

The functional acetyl group originates from Lys(Ac). By the reaction catalyzed by HDACs, acetyl group is released from ϵ -acetylated lysine moieties. In the first step, HDAC releases the acetyl group from lysine of the substrate. In the second step, the deacetylated peptide, now with unprotected lysine residue, is recognized by trypsin and subsequently cleaved to release 7-amino-4-methylcoumarin (AMC). This protocol is described by Wegener D *et al.* (Chemistry & Biology, 2003).

REFeree 1

What ingredient is the substrate for pentanoate fermentation?

AUTHOR RESPONSE

We thank the reviewer for pointing this out. The fecal concentrations of pentanoate are very similar to that of isopentanoate (isovalerate) and isobutyrate (all three of them are much lower than that of acetate, propionate and butyrate). It has been shown that in the intestine of humans and pigs, the most common dietary protein-derived fermentation products are ammonia, indole and branched-chain fat acids (BCFAs) isopentanoate and isobutyrate (Jha R. and Berrocoso J.D., *Animal*, 2015). Further, one study has shown that the high protein fermentation activity in the colon of dogs fed high protein diet (HPD) led to increased production of pentanoate suggesting a dietary protein-dependent dietary source for this SCFA (Nery J. *et al.*, *Journal of animal science*, 2012). Further in depth studies are needed in the future to clarify the exact ingredient that acts as substrate for pentanoate. In the follow-up study, we are planning to investigate the impact of HPD on the generation of various SCFAs.

REFeree 1

In Fig. 2d (and others, too), the signature Pentanoate, 2-DG on the upper region of the graph should be located below the graph.

AUTHOR RESPONSE

We have changed the signature of the figures (Fig. 2d, Fig. 3d and e, Fig. 3g, Fig. 3j, Fig. 5e, Fig. 6f and Fig. 6g) in the revised manuscript following the reviewer's advice. We hope the reviewer now supports the publication of this manuscript.

REVIEWERS' COMMENTS:

Reviewer #1 (Remarks to the Author):

The authors have adequately answered all my remarks. I congratulate them on a very interesting study.